# Protein Profiling of Serum Extracellular Vesicles Reveals Qualitative and Quantitative Differences after Differential Ultracentrifugation and ExoQuick™ Isolation

**DOI:** 10.3390/jcm9051429

**Published:** 2020-05-12

**Authors:** Timo Gemoll, Svitlana Rozanova, Christian Röder, Sonja Hartwig, Holger Kalthoff, Stefan Lehr, Abdou ElSharawy, Jens Habermann

**Affiliations:** 1Section for Translational Surgical Oncology & Biobanking, Department of Surgery, University of Lübeck and University Hospital Schleswig-Holstein, 23562 Lübeck, Germany; sv.rosanova@gmail.com (S.R.); jens.habermann@uni-luebeck.de (J.H.); 2Institute for Experimental Cancer Research, University of Kiel, 24105 Kiel, Germany; c.roeder@email.uni-kiel.de (C.R.); hkalthoff@email.uni-kiel.de (H.K.); 3Institute for Clinical Biochemistry and Pathobiochemistry, German Diabetes Center at the Heinrich-Heine-University Düsseldorf, Leibniz Center for Diabetes Research, 40225 Düsseldorf, Germany; sonja.hartwig@ddz.uni-duesseldorf.de (S.H.); stefan.lehr@ddz.uni-duesseldorf.de (S.L.); 4German Center for Diabetes Research (DZD), 85764 München-Neuherberg, Germany; 5Institute of Clinical Molecular Biology, Center of Molecular Sciences, University of Kiel, 24118 Kiel, Germany; a.sharawy@mucosa.de; 6Division of Biochemistry, Chemistry Department, Faculty of Sciences, Damietta University, New Damietta City 34511, Egypt; 7Interdisciplinary Center for Biobanking-Lübeck (ICB-L), University of Lübeck, 23562 Lübeck, Germany

**Keywords:** extracellular vesicles, isolation, inflammatory bowel disease, colorectal cancer, proteomics

## Abstract

Solid tumor biopsies are the current standard for precision medicine. However, the procedure is invasive and not always feasible. In contrast, liquid biopsies, such as serum enriched for extracellular vesicles (EVs) represent a non-invasive source of cancer biomarkers. In this study, we compared two EV isolation methods in the context of the protein biomarker detection in inflammatory bowel disease (IBD) and colorectal cancer (CRC). Using serum samples of a healthy cohort as well as CRC and IBD patients, EVs were isolated by ultracentrifugation and ExoQuick™ in parallel. EV associated protein profiles were compared by multiplex-fluorescence two-dimensional difference gel electrophoresis (2D-DIGE) and subsequent identification by mass spectrometry. Validation of gelsolin (GSN) was performed using fluorescence-quantitative western blot. 2D-DIGE resolved 936 protein spots in all serum-enriched EVs isolated by ultracentrifugation or ExoQuick™. Hereof, 93 spots were differently expressed between isolation approaches. Higher levels of GSN in EVs obtained with ExoQuick™ compared to ultracentrifugation were confirmed by western blot (*p* = 0.0006). Although patient groups were distinguishable after both EV isolation approaches, sample preparation strongly influences EVs’ protein profile and thus impacts on inter-study reproducibility, biomarker identification and validation. The results stress the need for strict SOPs in EV research before clinical implementation can be reached.

## 1. Introduction

Colorectal cancer (CRC) is the fourth leading cause of cancer death worldwide [1]. About 1–2% of CRCs are associated with inflammatory bowel disease (IBD) which comprises a group of chronic inflammatory disorders of the colon and small intestine. IBD-associated dysplasia is an important marker for increased CRC risk [2]. However, histopathology diagnostics of IBD-associated mucosa biopsies is clearly hampered and less reproducible among experts [3]. Thus, a non-invasive, clinically reliable screening program requires disease-specific biomarkers that accurately detect IBD, precancerous colorectal neoplasia and CRC at earliest stages.

Along with circulating tumor cells and circulating cell-free DNA, extracellular vesicles (EVs) are considered to be a promising source for liquid biopsy-based biomarker discovery allowing non-invasive screening, diagnostics, therapy guidance and repeated sampling for disease monitoring [4]. EVs are membrane-bound particles, secreted in vivo by cells into the extracellular environment and are detected in various biological fluids such as plasma, urine, saliva, ascites and bronchoalveolar lavage fluid [5]. EVs are classified into three groups: apoptotic bodies (1000–5000 nm), microvesicles (200–1000 nm) as well as exosomes (30–150 nm) and are found to play key roles in physiological and pathological events, e.g., intracellular communication [6,7], cell signaling [8], immune response [9,10] and carcinogenesis [5,11].

In terms of colorectal diseases, EVs appear to be involved in IBD pathogenesis [12] as well as CRC progression [13]. Depending on their cellular origin, EVs carry a selectively packaged cargo of DNA, mRNA, miRNA, lipids, metabolites and proteins [14]. The latest mediate the greater part of biological events in cells and are considered as promising biomarker source for various pathological states [13,15,16,17]. The main challenges for EV’s clinical application are the complex isolation, low level of standardized pre-processing, and high potential of pre-analytical bias. Examples are the choice of anticoagulant, the incubation time between blood collection and centrifugation as well as sample storage conditions that are all crucial for EVs’ quantity and quality [18]. In this context, the isolation process of EVs from blood is the most sensitive issue among all pre-analytical parameters influencing EVs yield, size, and RNA concentration [19,20].

Thus, the aim of our study was to compare and validate two different EV isolation approaches. Although many isolation protocols are published and available (e.g., ultrafiltration, size exclusion chromatography, affinity-based capturing), we decided to study ultracentrifugation as the most commonly used “gold standard” against the easy to use polymer-based ExoQuick™ precipitation kit. Next to the possible clinical implementation without laborious work, the latter has been additionally selected to avoid any contamination with antibodies, significant dilutions of the final sample and other confounding factors like viscosity. Furthermore, both isolation workflows were used to evaluate the potential for detecting differentially expressed proteins as biomarkers discerning healthy controls, IBD, and CRC.

## 2. Materials and Methods

### 2.1. Isolation of EVs

Studies were approved by the local Ethics Committees at the University of Lübeck (No. 07-124) and the Medical Faculty of the Christian-Albrechts-University of Kiel (A110/99). After signed informed consent, peripheral blood of clinical controls with neither oncological nor inflammatory bowel disease (n = 12) and patients with either IBD (n = 18) or sporadic colorectal cancer (n = 18) were collected based on standard operation procedures (SOPs) using a hospital-based biobank infrastructure (Appendix A). Serum was obtained from clotted venous blood samples by centrifugation at 2000× *g* for 10 min and stored either at −80 °C or in the gas-phase of liquid nitrogen at a temperature of −160 °C.

To ensure sufficient protein amounts of EVs, serum samples were pooled by mixing equal volumes of six individuals (1 mL) from the same group per pool (Appendix A). The pools were divided into two aliquots that were used for EV isolation by ultracentrifugation (2 × 250 µL) or a commercially available, polymer-based precipitation kit (500 µL, ExoQuick™, System Biosciences, Palo Alto, CA, USA). For differential ultracentrifugation, the serum samples were pre-centrifuged at 13,000× *g* for 10 min and at 100,000× *g* for 60 min at 4 °C. For EV polymer-based precipitation by the ExoQuick™ kit (System Biosciences, Palo Alto, CA, USA), the isolation was carried out according to the manufacturer’s protocol. EV pellets and supernatants obtained by both methods were stored at −80 °C until further analysis.

### 2.2. EVs Protein Enrichment and Purification

EV pellets from ultracentrifugation and ExoQuick™ precipitation were resuspended in 1.4 mL 1x PBS buffer. To reduce the concentration of abundant serum proteins, samples were treated with the ProteoMiner™ Protein Enrichment Kit (Bio-Rad Laboratories, Hercules, CA, USA). Each sample (150 µL) with and without ProteoMiner™ enrichment was diluted with 150 µL of sample preparation buffer [8 M urea, 4% (*w*/*v*) CHAPS, 2% (*v*/*v*) carrier ampholytes (pH 4-7), 40 mM DTT]. To achieve complete lysis, samples were incubated for 3h at room temperature including three integrated freeze-thaw cycles in liquid nitrogen every 60 min. Disturbing ionic contaminants such as detergents, lipids, and phenolic compounds, protein samples were treated with ReadyPrep™ 2-D Cleanup Kit (Bio-Rad Laboratories) as specified by the manufacturer.

### 2.3. Two-Dimensional Multiplex Fluorescence Gel Electrophoresis (2D-DIGE)

Total protein concentration in samples was determined using the fluorescence-based EZQ™ Protein Quantitation Kit (Life Technologies, Carlsbad, CA, USA). A total of 50 µg of each protein sample and a pooled internal standard were labelled with the fluorescence-based Refraction-2D™ Labeling Kit (NH DyeAGNOSTICS, Halle, Germany) according to the manufacturer’s protocol. One hundered and fifty µg of protein per gel (2 × 50 µg sample plus 50 µg internal standard) were diluted with rehydration sample buffer to a final volume of 450 µL and applied to immobilized pH gradient (IPG) gel strips with a pH range 4-7 by means of an active sample in-gel rehydration approach under gentle voltage (50 V, 6 h and 60 V, 11 h). Isoelectric focusing (IEF) was carried out in a Protean^®^ i12TM IEF cell (Bio-Rad Laboratories) at 20 °C reaching approximately 57,700 Vh. After IEF, the IPG strips were immediately equilibrated for 2 × 15 min in a premade buffer system containing tris-tricine/SDS (pH 6.9) (Buffer Kit for 2D HPE™ Gels, SERVA Electrophoresis, Heidelberg, Germany). To reduce S-S bonds and alkylate free thiols, 2% (*w*/*v*) DTT was included to the buffer in the first and 2.5% (*w*/*v*) IAA in the second equilibration step. The horizontal second dimension (HPE™ FlatTop Tower, SERVA Electrophoresis) was carried out by SDS-PAGE on precast plastic-backed 12.5% acrylamide gels (2DHPE™ Large Gel NF 12.5% Kit, 0.65 × 200 × 255 mm, SERVA Electrophoresis). Gel-imaging, gel analysis and mass spectrometric analysis for spot identification were carried out as described previously [21].

### 2.4. Multiplex Fluorescence-Based Western Blot Analysis

The protein gelsolin (GSN) was identified by 2D-DIGE with subsequent MS and further validated by quantitative western blot [22]. Briefly, 5 µg of each protein sample and a pooled internal standard, used to normalize potential gel-to-gel variations, were labeled with T-Rex Protein Labeling Kit (NH DyeAGNOSTICS). EVs proteins were separated on precast 4–15% polyacrylamide gels (Criterion™ TGX™ Protein Gel, Bio-Rad Laboratories). After SDS-PAGE, separated proteins were electroblotted onto a PVDF membrane (Immobilon^®^-FL PVDF, 0.45 µm, Merck KGaA, Darmstadt, Germany) using a Trans-Blot^®^ Turbo™ Transfer System (Bio-Rad Laboratories). The membrane was blocked and incubated with primary antibodies against gelsolin (GSN, 1:1000 rabbit monoclonal antibody, clone D9W8Y, Cell Signaling, Danvers, MA, USA) in 2% blocking buffer at 4 °C overnight. Afterwards, blots were incubated for 1 h at room temperature with Cy3-conjugated goat-anti-rabbit secondary antibodies (Amersham ECL™Plex CyDye-Conjugated Antibodies, GE Healthcare, Chicago, IL, USA) diluted 1:2500 in 2% blocking buffer. Final protein fluorescence visualization was carried out with a Typhoon™ FLA 9000 laser scanner (GE Healthcare). Densitometric analyses of loaded total protein and antibody-targeted protein bands were performed using the ImageQuant™ TL software (GE Healthcare). Each specific antibody-targeted protein band (Cy3 channel detection) was first normalized against the loaded total protein volume value (Cy5 channel detection) of the corresponding EV sample (Cy3/Cy5 ratio).

## 3. Results

The aim of our study was to compare and validate two different EV isolation approaches (ultracentrifugation vs. ExoQuick™) on the proteomic level by two-dimensional gel electrophoresis and mass spectrometry. While ultracentrifugation is usually regarded as the “golden standard” for EVs’ isolation and primarily based on the size of extracellular particles, the polymer-based isolation by the ExoQuick™ kit uses a polymer solution which creates a polymer network allowing the separation of EVs by low-speed centrifugation.

### 3.1. EV Protein Profile Depending on Isolation Method

While 2D-DIGE resolved 750 protein spots in gels of EV pellets without ProteoMiner™ enrichment, 936 protein spots were detected in ProteoMiner™ enriched EV pellets isolated by ultracentrifugation or ExoQuick™ (Appendix A). Due to the higher resolution, all further analyses were carried out using ProteoMiner™-enriched EVs. In order to check for possible contamination of EVs with serum derived proteins, protein profiles obtained for isolated EVs and the corresponding serum supernatants were compared. Unsupervised cluster analysis of all detected 936 protein spots resulted in distinct group clustering of EV proteins and supernatant proteins for both, being isolated by ultracentrifugation (Figure 1A) or ExoQuick™ precipitation (Figure 1B).

Hierarchical cluster analysis of all 936 protein spots revealed two major clusters for both ultracentrifugation and ExoQuick™ isolation (Figure 1 and Appendix A). 226 EV protein spots overlapped between two isolation approaches. Protein spots determined for either ultracentrifugation or ExoQuick™ as supernatant-relevant were considered to be serum contaminants and were thus excluded from the following analysis.

Subsequent comparison between both isolation protocols revealed in total 93 proteoforms that were differently expressed (*p* < 0.05 & *q* < 0.05) resulting in a distinct group clustering by a supervised PCA (Figure 2). While the levels of 56 out of 93 protein spots were significantly increased in ultracentrifuged EV pellets, 37 protein spots presented higher protein levels in ExoQuick™-EV pellets.

### 3.2. Mass Spectrometry & Pathway Analyses

All 93 EV pellet protein spots found to be significantly different between ultracentrifugation and ExoQuick™ were picked for subsequent mass spectrometric analysis. Hereof, 39 (42%) protein spots were identified by Mascot database representing 21 protein identities (Table 1).

Except for immunoglobulin mu heavy chain disease protein (MUCB) and zinc finger protein 705A (ZNF705A), all identities were reported as human exosome proteins. Overall analysis of identified protein spots revealed that 25 (AFM, A2M, C4BPA, CFH, HP, IGKC, MUCB, ITIH4, KRT2, PON1, ZNF705A) and 14 (SERPINC1, CD5L, C1R, C6, IGHM, GC) spots to be significantly higher expressed (*p* < 0.05) in ultracentrifuged and ExoQuick EVs, respectively (Table 2).

### 3.3. Potential Diagnostic Power of EV Proteins

The two-group comparison between clinical controls and IBD samples revealed 31 spots for ultracentrifugation and 18 spots for ExoQuick™ isolation to be differentially expressed. The analysis of clinical controls and CRC samples showed 62 and 21 differentially regulated proteoforms for ultracentrifugation and ExoQuick™, respectively, with an overlap of eight protein spots. Two distinct proteins (GSN and A2M) were subsequently identified. IBD and CRC samples were separated by 32 and 26 differently expressed spots after ultracentrifugation or ExoQuick™ with an overlap of seven protein spots in total. Independent of the EV isolation method, PCA analyses revealed distinct between-group separations (Figure 3).

### 3.4. Validation of Target Proteins by Fluorescence-Based Western Blot

Based on the 2-DIGE evaluation of different isolation approaches, availability of antibodies, exosome-specific molecular function, GSN was selected for downstream validation by multiplex fluorescence-based Western blot analysis. Gelsolin protein level was significantly higher in samples obtained with ExoQuick™ compared to ultracentrifugation (*p* = 0.0006) (Figure 4).

However and e.g., due to small samples size as well as specific isoform detection of the used antibody, western blot did not show any significant differences between controls, IBDs and CRCs groups (Appendix A).

## 4. Discussion

Although cancer biomarker discovery based on extracellular vesicles (EVs) has become a popular research field, it is highly restricted by challenges in EVs isolation and characterization. Current isolation methods differ in terms of extraction efficiency as well as quality and purity of the obtained EVs [19,23]. Depending on the applied isolation approach, the obtained mixture of EVs’ subpopulations may strongly differ thus affecting sample quality in terms of macromolecule content. In turn, such variations in sample quality is crucial for the reliability and validity of research findings. Moreover, a wide diversity of existing EVs isolation protocols may strongly interfere with verification, comparison, and analysis of the data obtained by different research teams. With the background that the acquisition of EV fractions still remains technically challenging [24], we compared two commonly used approaches for EV isolation on the proteomic level using clinical cohorts of healthy volunteers, clinical controls and patients with either IBD or CRC.

Ultracentrifugation is usually regarded as the “golden standard” for EVs’ isolation. However, the outcome for this approach is highly dependent on centrifugation time and speed, type of rotor as well as temperature [25]. Moreover, time consuming ultracentrifugation induces the formation of EV aggregates composed of a mixture of EVs of various phenotypes and morphologies [26]. On the other hand, polymer-based isolation using ExoQuick™ isolation kit is another widely used technique. It is based on mixing the sample with a polymer solution which, at specific salt conditions and temperature, creates a polymer network allowing the isolation of EVs by low-speed centrifugation [27]. This method is simple, fast, and requires as much as 250 µl of serum. Yet, it is also cost-intensive. It allows for a greater yield of the EVs in comparison with ultracentrifugation but with a lower purity and thus decreased EV specificity [19]. ExoQuick™ samples contain a high portion of salts, polymer and other contaminants (including lipoproteins) especially from serum samples [23,28]. Additionally, incompatibilities for exosome specific protein marker expression due to sample preparation were reported after assessing EV samples with downstream validation approaches [29].

Quantitative differences between serum-derived EVs obtained by ultracentrifugation and ExoQuick™ were observed in a distinct PCA group clustering based on 93 (~10%) significant different expressed spots. A total of 39 EV protein spots were identified which are generally involved in immune response, acute inflammatory response, defense response, cytoskeleton organization, and vitamin transport processes. Interestingly and despite the differences in protein expression, both isolation approaches allowed distinguishing healthy and clinical controls from patients with IBD and CRC. Inter-group comparison between healthy controls and patients with IBD and sporadic CRC identified ten proteins from the ultracentrifugation (i.e., AFM, A2M, C4BPA, C4B, C6, GSN, IGHM/MUCB, ITIH4, GC, HP) and seven proteins from the ExoQuick™ group (i.e., A2M, C4B, GSN, IGHA1, IGHA2, IGHM/MUCB, SERPINC1) to be differentially expressed (*p* < 0.05). Four of those, namely A2M, C4BPA, IGHM/MUCB, and GSN showed an overlap belonging to both groups.

While A2M acts as a protease inhibitor during blood coagulation and platelet degranulation, C4BPA binds as a cofactor in the complement activation pathway and IGHM/MUCB serves as a receptor during humoral immunity. A2M, C4BP and IGHM/MUCB coact in biological processes of primary defense mechanisms, e.g. regulation of the complement cascade as well as immune response. In line with our results, it has been shown that decreased protein levels in cancerous samples of A2M, C4BP and IGHM/MUCB disturb the otherwise protective role of these proteins and may abet tumor development [30,31,32]. Although further studies are needed to evaluate the correlation with patients’ individual characteristics, the protein content of cancer-derived EVs could thus support minimal-invasive cancer diagnosis and prognosis.

GSN was further selected for western blot validation which confirmed a higher protein level after ExoQuick™ isolation compared to ultracentrifugation preparation. GSN is an actin-binding protein that regulates cell growth and motility as well as maintains the integrity of cytoskeletal structure. GSN have been demonstrated in various tumor cells and tissues [33,34], including colorectal adenocarcinoma tissue [35], and was exclusively and specifically found within exosomes and not in a large size EVs populations like microvesicles and apoptotic bodies [36]. As GSN is further widely debated as biomarker candidate in different cancers with evidence supporting its contradictory involvement in both tumor suppression as well as malignant progression [32,35], the reported data stress the need for strict SOPs research. Both, GSN expression and the global EV protein profiles, highlight the impact of EVs isolation approaches and thus the challenging use of EVs for the translation into clinical applications. In line, Abramowicz et al. [37] reviewed that the choice of isolation methods significantly influence mass spectrometric results and data interpretation.

In conclusion, we demonstrated that EV protein expression is highly dependent on the isolation approach. On the proteomic level, ultracentrifugation and ExoQuick™ seems to be two complementary approaches allowing the detection of different proteoforms with different abundance and purity levels. Thereby, engagement of highly standardized operating procedure for EVs isolation, handling and analyzing in combination with an increased transparency for data reporting are needed for implementation of non-invasive EV-based biopsies for cancer diagnostic and prognostic.

## Figures and Tables

**Figure 1 jcm-09-01429-f001:**
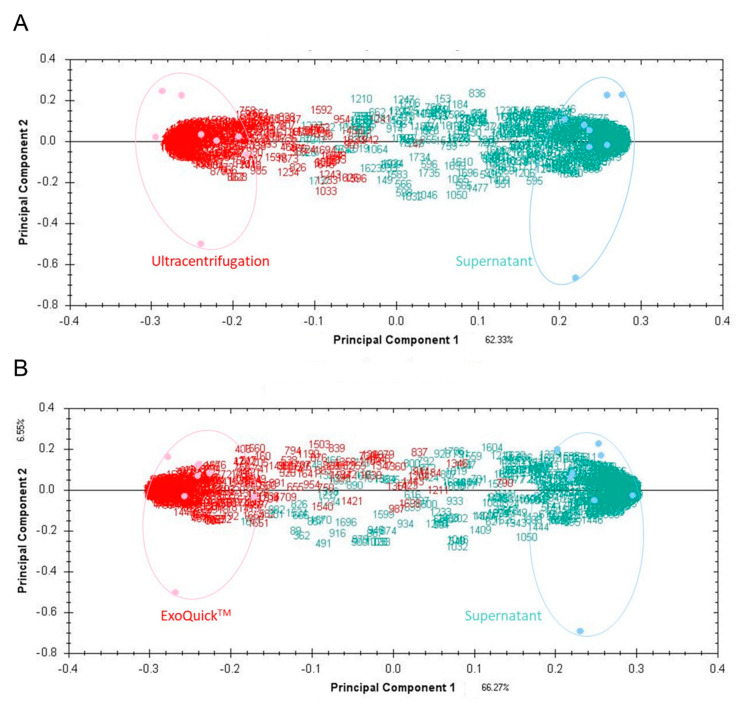
Unsupervised PCA plots based on all 936 total protein spots detected between EV pellets and corresponding serum supernatants isolated by (**A**) ultracentrifugation or (**B**) ExoQuick™. Red, proteins with higher expression in Ultracentrifugation/ExoQuick™; Green, proteins with higher expression in the supernatant.

**Figure 2 jcm-09-01429-f002:**
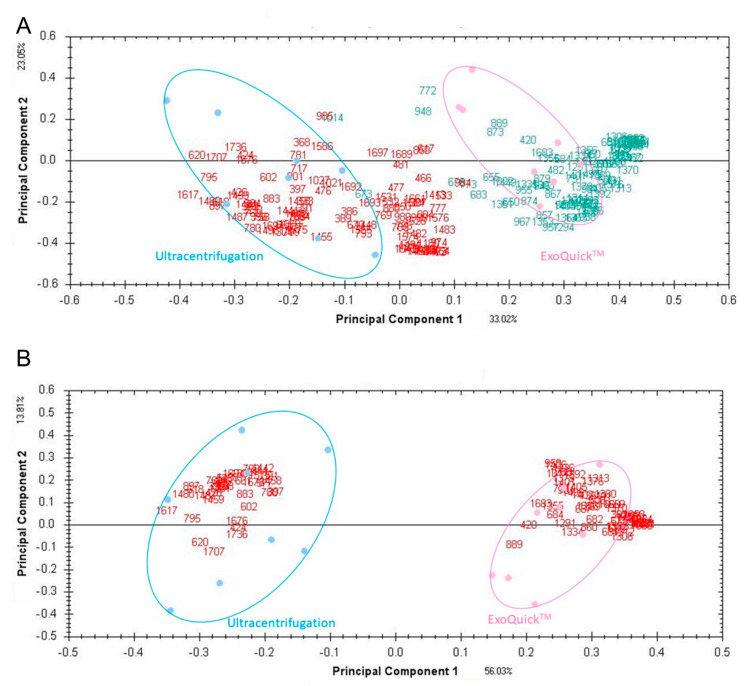
Supervised PCA plots based on 226 (**A**) and 93 (**B**) significant spots obtained by ultracentrifugation or ExoQuick. Red, proteins with higher expression isolated by Ultracentrifugation; Green, proteins with higher expression isolated by ExoQuick™.

**Figure 3 jcm-09-01429-f003:**
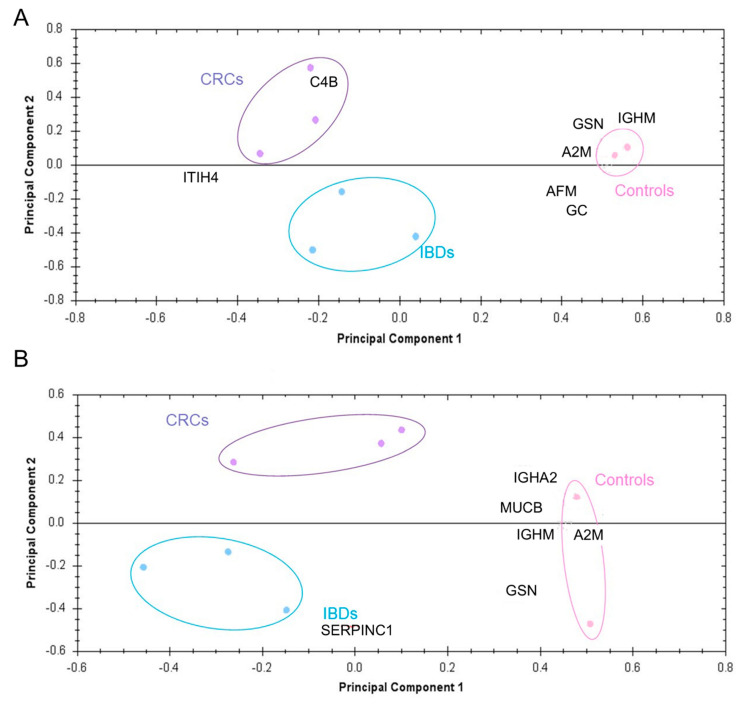
PCA plots based on identified proteins between clinical control (in pink), IBD (in blue) and CRC (purple) EV pellet samples isolated by (**A**) ultracentrifugation (n = 7) and (**B**) ExoQuick™ (n = 6).

**Figure 4 jcm-09-01429-f004:**
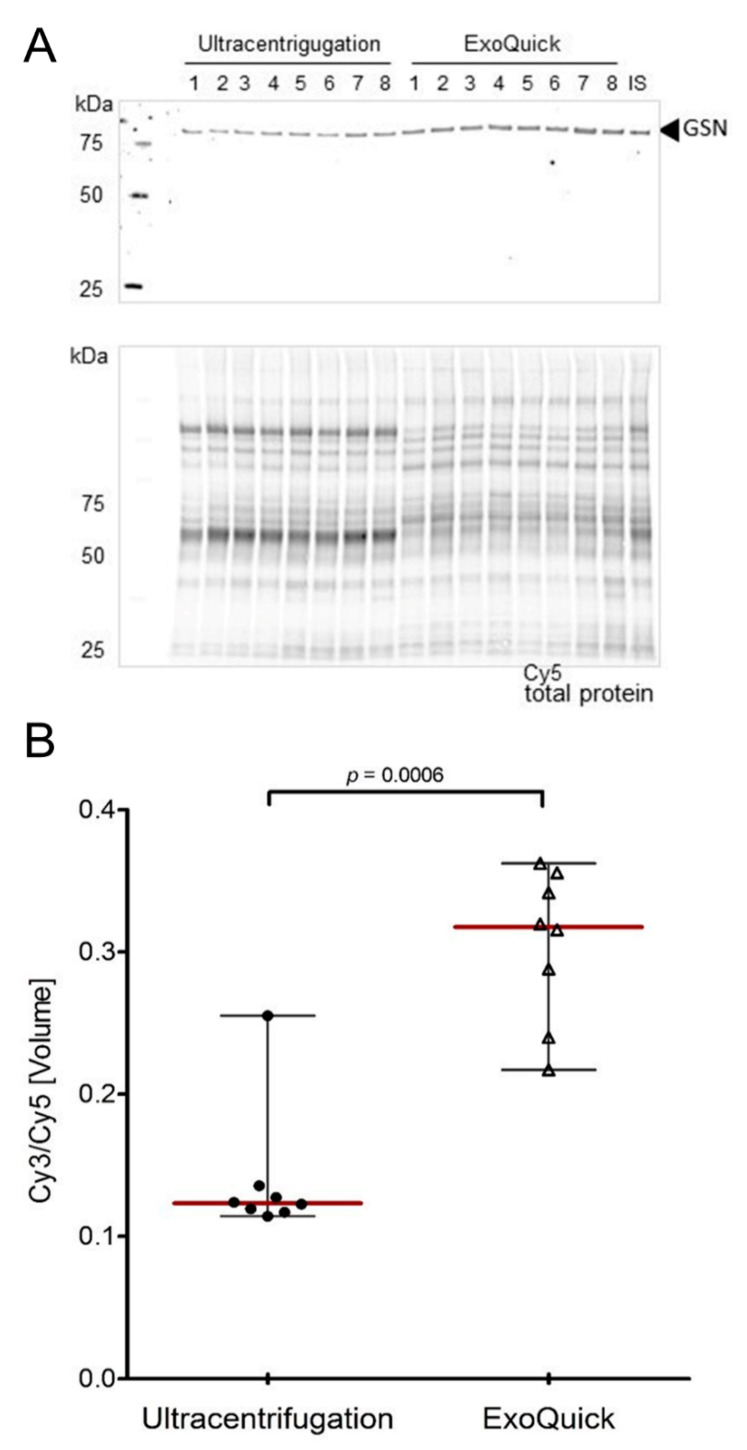
Multiplex fluorescent-based Western blot analysis of EVs isolated with ultracentrifugation and ExoQuick™. Specific antibody-targeted protein bands of Gelsolin were detected by Cy3-labeled secondary antibody (**A**). Cy5 total protein signals within each lane were used for normalization (Cy3/Cy5 ratio). Based on an internal standard (IS), adjusted relative protein level calculation and statistical analysis were performed (**B**).

**Table 1 jcm-09-01429-t001:** Number of differently expressed as well as identified spots in two- and three- group comparison of EV pellet samples isolated by ultracentrifugation and ExoQuick™.

	Ultracentrifugation (UC)		ExoQuick™		Overlapped Protein Spots	
	Differentially Expressed (*p* < 0.05) Spots	Identified Spots	Differentially Expressed (*p* < 0.05) Spots	Identified Spots	Total Number	Identified Spots
Controls vs IBDs	31	9	18	9	7	5
Controls vs. CRC	62	16	21	4	8	3
IBDs vs. CRCs	14	5	17	5	7	3
Controls vs. IBDs vs. CRCs	32	10	26	11	8	6

**Table 2 jcm-09-01429-t002:** Summary of identified EV proteins using MALDI-TOF/TOF-MS.

Spot No.	Protein Identity	Gene *Symbol*	Accession No.	MW [kDa]^a^	pI^a^	MW [kDa]^b^	pI^b^	Mascot Score (MS/MS)	Sequence Coverage [%]	UC vs. Exo (Pellets)	Ctrl. vs. IBD (UC)	Ctrl. vs. CRC (UC)	IBD. vs. CRC (UC)	Ctrl. vs. IBD vs. CRC (UC)	Ctrl. vs. IBD (Exo)	Ctrl. vs. CRC (Exo)	IBD. vs. CRC (Exo)	Ctrl. vs. IBD vs. CRC (Exo)
1	Afamin	AFM	P43652	139	5.19	69.0	5.6	61.9	13.7	↓		↓		↓				
2	Alpha-2-macroglobulin	A2M	P01023	186	6.24	163.2	6.0	145.4	3.5	↓	↓	↓		↓	↓	↓		↓
3		A2M		186	6.19			86.2	24.2	↓								
4		A2M		186	6.28			131.0	22.7	↓	↓	↓		↓	↓			↓
5		A2M		186	6.32			89.4	1.9	↓	↓	↓		↓	↓			↓
6		A2M		186	6.14			86.4	1.9	↓		↓						
7		A2M		185	6.41			85.6	19.8	↓	↓	↓		↓	↓			↓
8	Antithrombin-III	SERPINC1	P01008	118	5.32	52.6	6.3	72.4	3.0	↑							↓	↓
9	C4b-binding protein alpha chain	C4BPA	P04003	124	6.55	67.0	7.9	97.0	34.2	↓								
10		C4BPA		71	5.22			127.0	35.3	↓		↓	↓					
11	CD5 antigen-like	CD5L	O43866	65	5.60	38.1	5.2	70.8	25.1	↑								
12	Complement C1r subcomponent	C1R	P00736	65	5.36	80.1	5.8	79.9	3.7	↑								
13	Complement C4-B	C4B	P0C0L5	57	4.29	192.6	6.9	72.0	0.7	↑		↑	↑	↑				
14		C4B		56	4.73			72.9	1.5	↓							↑	
15	Complement component C6	C6	P13671	170	6.46	104.7	6.4	210	37.4	↑			↓					
16		C6		169	6.53			138.0	33.4	↑			↓					
3	Complement factor H	CFH	P08603	186	6.19	139.0	6.2	56.1	30.8	↓								
17		CFH		186	6.01			176.0	40.1	↓								
18	Gelsolin	GSN	P06396	141	6.32	85.6	5.9	70.2	2.8	↑		↓				↓		↓
19		GSN		140	6.34			59.1	22.0	↑	↓	↓		↓		↓		↓
20		GSN		141	6.23			92.0	19.6	↑						↓		↓
14	Haptoglobin	HP	P00738	56	4.73	45.2	6.1	90.6	30.5	↓		↑						
21	Ig alpha-1 chain C region	IGHA1	P01876	80	6.29	37.6	6.1	118.0	47.9	↓								
22		IGHA1		107	5.61			97.0	40.8	↓								
23		IGHA1		116	5.30			64.8	33.7	↑					↓		↓	
24		IGHA1		108	6.62			61.9	46.5	↑								
25		IGHA1		105	6.02	37.6	6.1	184.0	56.9	↓								
26		IGHA1		105	6.47			60.6	18.7	↓								
27	Ig alpha-2 chain C region	IGHA2	P01877	107	5.74	36.5	5.7	244.0	8.5	↓					↓		↑	↓
25		IGHA2		105	6.02			118.0	40.3	↓								
28		IGHA2		104	6.20			92.1	5.0	↓								
21		IGHA2		80	6.29			61.9	36.2	↓								
29		IGHA2		106	5.80			80.9	22.4	↓					↓			
22		IGHA2		107	5.61			66.2	31.5	↓								
23		IGHA2		116	5.30			71.4	5.0	↑					↓		↓	
8		IGHA2		118	5.32			87.3	5.0	↑								
30	Ig kappa chain C region	IGKC	P01834	24	6.39	11.6	5.5	89.8	32.1	↓								
31		IGKC		24	6.60	11.6	5.5	92.6	34.9	↓								
32	Ig mu chain C region	IGHM	P01871	60	6.26	49.3	6.4	57.7	13.3	↑	↓	↓						
33		IGHM		129	6.31			105.5	9.7	↑	↓	↓		↓	↓			↓
34	Ig mu heavy chain disease protein*	MUCB*	P04220*	67	6.28	43.0	5.0	157.4	3.3	↓	↓	↓						
35		MUCB*		141	6.55			81.0	2.8	↓		↓						↓
36	Inter-alpha-trypsin inhibitor heavy chain H4	ITIH4	Q14624	168	5.11	103.3	6.5	76.1	17.8	↓	↑	↑		↑				
37	Keratin, type II cytoskeletal 2	KRT2	P35908	27	5.15	65.4	8.9	68.8	35.7	↓								
14	Serum paraoxonase/arylesterase 1	PON1	P27169	56	4.73	39.7	5.0	94.9	4.5	↓		↑						
38	Vitamin D-binding protein	GC	P02774	90	5.29	52.9	5.3	203.0	62.2	↑		↓	↓	↓				
39	Zinc finger protein 705A	ZNF705A	Q6ZN79	41	6.43	34.7	10.3	57.3	26.3	↓								

39 protein spots significantly different expressed (*t*-test, *p* < 0.05) between EV samples, obtained with ultracentrifugation (UC) and ExoQuick™ (Exo). Arrows indicate significant protein down- (↓) or upregulation (↑) in the latter sample of each comparison (*t*-test, *p* < 0.05 and 1-way ANOVA, *p* < 0.05). ^a^ observed values, ^b^ theoretical values (UniProt database). Contr., control; CRC, colorectal cancer; IBD, inflammatory Bowel disease. * replaced by UniProt for P01871.

## Data Availability

The data that support the findings of this study are available on request from the corresponding author. The data are not publicly available due to privacy or ethical restrictions.

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
