# Peer review of "Protein Profiling of Serum Extracellular Vesicles Reveals Qualitative and Quantitative Differences after Differential Ultracentrifugation and ExoQuick™ Isolation"

_jcm, 2020, doi:10.3390/jcm9051429_

Round 1
Reviewer 1 Report
In this paper a comparative analysis of two methods to enrich EV from plasma of healthy Controls, inflammatory bowel disease (IBD) and colorectal cancer (CRC) patients was performed with ultracentrifugation and ExoQuick precipitation, The work is interesting and well performed however similar papers has been already published in this issue such as Bioengineering 2019, 6, 8; doi:10.3390/bioengineering6010008, where a comparison of three methods to enrich EV from plasma was performed with ultracentrifugation, ExoQuick and also Total Exosome Isolation kit. Although this point the results may help on having more information about these techniques.
Minor revision should carry about in the paper;
- In Line5 ; Correct the typo error; In terms of colorectal diseases, EVs appear to be involved in in IBD pathogenesis
- Table 1 and figure 1 are more appropriate for supporting information
- Before explaining results (section 3.1) it is necessary that the authors introduce the compared technique; how both techniques works and how these techniques are able to separate EV from the rest of cells.
- The section 3,1 should starts introducing what they are doing first; the comparison of protein enrichment not directly the results obtained.
- In the numbering of the pictures in figure 1 use capital letters in the left upper corner as in the rest of figures
- In the introduction are not considered other technique similar to ultracentrifugation and ExoQuick that should be introduced and explain why has been choose ExoQuick as the best candidate to compare with the standard.
- In figures 2, 3 and 4 remove the graph titles and the icons appearing on the axes.
- Explain the term Unsupervised/supervised PCA plots in the results section
- Although ExoQuick is able to separate more proteins the standard is able to reveal 31 vs 18 spots for ExoQuick™ isolation to be differentially expressed in Controls and IBD. Also, IBD and CRC samples were separated by 32 and 26 differently expressed spots after ultracentrifugation or ExoQuick™ Do not make this technique a better technique? Introduce this point in the discussion
- Why with western blot is not possible to detect differences? Discusses it
- The authors said that ultracentrifugation is highly dependent on centrifugation time and speed, type of rotor as well as temperature, but ExoQuickTM also depend on centrifugation, it has been tested it dependence on centrifugation time and speed, type of rotor?
Author Response
Reviewer 1:
In this paper a comparative analysis of two methods to enrich EV from plasma of healthy Controls, inflammatory bowel disease (IBD) and colorectal cancer (CRC) patients was performed with ultracentrifugation and ExoQuick precipitation, The work is interesting and well performed however similar papers has been already published in this issue such as Bioengineering 2019, 6, 8; doi:10.3390/bioengineering6010008, where a comparison of three methods to enrich EV from plasma was performed with ultracentrifugation, ExoQuick and also Total Exosome Isolation kit. Although this point the results may help on having more information about these techniques.
We thank the reviewer for the evaluation of the manuscript. We agree that some studies have been published about the EV isolation methods and comparisons, however, to our knowledge this is the first study that a) compared EV isolation methods on the global proteome level (Top-down, untargeted) and b) developed a new combination including the ProteoMiner kit to enhance the detection sensitivity. We hope that the reviewer will find these comments convincing and thus the manuscript suitable for publication.
Minor revision should carry about in the paper;
- In Line5 ; Correct the typo error; In terms of colorectal diseases, EVs appear to be involved in in IBD pathogenesis
We adapted the manuscript accordingly.
- Table 1 and figure 1 are more appropriate for supporting information
We adapted the manuscript accordingly and transferred table 1 and figure 1 to the supplemental data.
- Before explaining results (section 3.1) it is necessary that the authors introduce the compared technique; how both techniques works and how these techniques are able to separate EV from the rest of cells.
- The section 3,1 should starts introducing what they are doing first; the comparison of protein enrichment not directly the results obtained.
We thank the reviewer for this important point and adapted the manuscript accordingly on page 4.
- In the numbering of the pictures in figure 1 use capital letters in the left upper corner as in the rest of figures
We adapted the manuscript accordingly.
- In the introduction are not considered other technique similar to ultracentrifugation and ExoQuick that should be introduced and explain why has been choose ExoQuick as the best candidate to compare with the standard.
We thank the reviewer for this very important point and added this information in the introduction on page 2.
- In figures 2, 3 and 4 remove the graph titles and the icons appearing on the axes.
We adapted the manuscript accordingly.
- Explain the term Unsupervised/supervised PCA plots in the results section
We adapted the manuscript accordingly on the pages 4-6 as well as in the figure caption of figure 1 & 2.
- Although ExoQuick is able to separate more proteins the standard is able to reveal 31 vs 18 spots for ExoQuick™ isolation to be differentially expressed in Controls and IBD. Also, IBD and CRC samples were separated by 32 and 26 differently expressed spots after ultracentrifugation or ExoQuick™ Do not make this technique a better technique? Introduce this point in the discussion
We thank the reviewer for this interesting point. However, we consider that small difference of significant protein spots detected by two-dimensional gel electrophoresis does not reflect the performance of the technique in terms of separation power. From our point of view, it is more straightforward that both methods act complementary.
As there is a likelihood to detect false-positive spots (missing specificity), which we now added on page 13, further validation studies including the EV specificity of the detected spots are warranted but are beyond the focus of the presented work.
- Why with western blot is not possible to detect differences? Discusses it
As the focus of the study presents the differences between EV isolation methods, we added an explanation for the missing validation in the disease groups in the results section on page 13.
- The authors said that ultracentrifugation is highly dependent on centrifugation time and speed, type of rotor as well as temperature, but ExoQuickTM also depend on centrifugation, it has been tested it dependence on centrifugation time and speed, type of rotor?
We thank the reviewer for this interesting point. We did not tested the centrifugation time and speed as well as the type of rotor for the ExoQuick™ kit in this study. However, this would be interesting study, we believe that the relatively low centrifugation speed used for the ExoQuick™ isolation is not that critical as the high speed of centrifugation in the ultracentrifugation method. Next, the ExoQuick™ method is more dependable on the solubility of the molecules in polymer solution.
As we are currently working on the combination of both methods (ExoQuick™ and ultracentrifugation), we will take this idea for future developments.
Reviewer 2 Report
Minor suggestions
Line 55 “In terms of colorectal diseases, EVs appear to be involved in in IBD pathogenesis [12] as well as”
Table 1. Patient cohort for 2D-DIGE protein profiling of extracellular vesicles. As it is in the present format, it does not clearly show the association of patients to the numbered pools or the pools to the disease groups. Although the text says that each pool is formed by 6 patients.
Line 81 “To ensure sufficient protein amounts of EVs, serum samples were pooled by mixing equal volumes of six individuals from the same group per pool (Table 1). Please specify the used volume of sample of each patient. Or the total amount of the pool sera.
Line 90 “EV pellets from ultracentrifugation and ExoQuickTM precipitation were resuspended in 1,400 μl (1.4 mL?)
Figure 4. PCA plots based on identified protein spots between clinical control (in pink), IBD (in blue) and CRC (purple) EV pellet samples isolated by (a) ultracentrifugation (n=10) and (b) ExoQuickTM (n=12). In this figure I do not see the same amount of spots as indicated in the legend. In my opinion, since the number of spot is low, it would be illustrative to name the protein corresponding to each spot in the figure.
Line 213 Moreover, a wide diversity of exciting? EVs isolation protocols may strongly interfere with
Major concerns:
In my opinion it is a well written manuscript and with technical quality but not very illustrative for the scientific community due to its too generic conclusions. The need to implement comparable protocols in liquid biopsy does not seem very novel. I think it could be improved with a more functional discussion of the 5 proteins that have been found in common in the two procedures and their ability to differentiate cancerous or precancerous processes in the terms of the study.
Author Response
Reviewer 2:
Minor suggestions
Line 55 “In terms of colorectal diseases, EVs appear to be involved in in IBD pathogenesis [12] as well as”
We adapted the manuscript accordingly.
Table 1. Patient cohort for 2D-DIGE protein profiling of extracellular vesicles. As it is in the present format, it does not clearly show the association of patients to the numbered pools or the pools to the disease groups. Although the text says that each pool is formed by 6 patients.
We reformat the table accordingly.
Line 81 “To ensure sufficient protein amounts of EVs, serum samples were pooled by mixing equal volumes of six individuals from the same group per pool (Table 1). Please specify the used volume of sample of each patient. Or the total amount of the pool sera.
We added the information on page 3 of the manuscript.
Line 90 “EV pellets from ultracentrifugation and ExoQuickTM precipitation were resuspended in 1,400 μl (1.4 mL?)
We adapted the manuscript accordingly.
Figure 4. PCA plots based on identified protein spots between clinical control (in pink), IBD (in blue) and CRC (purple) EV pellet samples isolated by (a) ultracentrifugation (n=10) and (b) ExoQuickTM (n=12). In this figure I do not see the same amount of spots as indicated in the legend. In my opinion, since the number of spot is low, it would be illustrative to name the protein corresponding to each spot in the figure.
Thank you for this valuable suggestion. We adapted the manuscript accordingly and presented the number of identified unique proteins which different to the number of significant protein spots and identified protein spots.
Line 213 Moreover, a wide diversity of exciting? EVs isolation protocols may strongly interfere with
We adapted the manuscript accordingly.
Major concerns:
In my opinion it is a well written manuscript and with technical quality but not very illustrative for the scientific community due to its too generic conclusions. The need to implement comparable protocols in liquid biopsy does not seem very novel. I think it could be improved with a more functional discussion of the 5 proteins that have been found in common in the two procedures and their ability to differentiate cancerous or precancerous processes in the terms of the study.
We thank the reviewer for this very important point. The main focus of the study was to compare two different isolation methods for subsequent extracellular vesicle (EV) characterization on the global proteome level including the detection of isoforms. However and additionally due to RNA analysis (not part of the paper), we were forced to pool samples in this initial profiling step in order to get sufficient RNA and protein material for the planned experiments. Furthermore, we decided to pool individual patients to exclude biological and individual differences between patients for a straight-forward experimental set-up of the isolation comparison. We suspected that without pooling patient samples a clear decision could not be drawn whether a protein is differentially regulated because of the patients’ entity or because of the isolation method. Knowing now that ExoQuick™ and ultracentrifugation for EV isolation are rather complementary (main result of this study), we have already started to isolate EVs of healthy controls and patients with IBD as well as CRC on the individual basis. We would be very happy to submit these results also to your journal as a continuation of our common goal to support patient care.
Round 2
Reviewer 2 Report
I think that the major concern has not been fulfilled
"Discussion could be improved with a more functional discussion of the 5 proteins that have been found in common in the two procedures and their ability to differentiate cancerous or precancerous processes in the terms of the study".
Author Response
Major concerns:
"Discussion could be improved with a more functional discussion of the 5 proteins that have been found in common in the two procedures and their ability to differentiate cancerous or precancerous processes in the terms of the study".
We thank the reviewer again for this valuable point. Although we still think that the focus of the study is the comparison of two different isolation methods for subsequent extracellular vesicle (EV) characterization on the global proteome level, we now added a functional discussion of detected proteins on page 13 and support the possibility to use proteins of EVs for minimal-invasive cancer diagnostics and prognostics. We hope that the adjustments are in line with the reviewers' view and that the manuscript is now suitable for publication.
Round 3
Reviewer 2 Report
I thanks the efforts of authors to complete the discussion section.